# Characteristics and Comparative Analysis of the Special-Structure (Non-Single-Circle) Mitochondrial Genome of *Capsicum pubescens* Ruiz & Pav

**DOI:** 10.3390/genes15020152

**Published:** 2024-01-24

**Authors:** Di Wu, Wenting Fu, Gaoling Fan, Dongfu Huang, Kangyun Wu, Yongfa Zhan, Xiangmin Tu, Jianwen He

**Affiliations:** 1Research Institute of Pepper, Guizhou Academy of Agricultural Science, Huaxi District, Guiyang 550025, China; 876865710@aliyun.com (D.W.); fu20210802@outlook.com (W.F.); fangaoling@aliyun.com (G.F.); dfh_881104@126.com (D.H.); wu_712@126.com (Y.Z.); gz2009txmzy@aliyun.com (X.T.); 2Key Laboratory of Crop Genetic Resources and Germplasm Innovation in Karst Mountain Areas, Ministry of Agriculture and Rural Affairs, Huaxi District, Guiyang 550025, China; swu522@126.com

**Keywords:** *Capsicum pubescens*, mitochondrial genome, gene clusters, selective pressure, phylogeny

## Abstract

Chilean peppers, cultivated from *Capsicum pubescens*, are globally renowned as popular vegetable and spice crops. *C. pubescens* belongs to the *Capsicum* L. (pepper) family and is one of the five pepper cultivars grown in China. In this study, we assembled and annotated the complete mt genome of *C. pubescens*. We investigated several aspects of its genome, including characteristics, codon usage, RNA editing sites, repeat sequences, selective pressure, gene clusters, and phylogenetic relationships. Furthermore, we compared it with other plant mt genomes. The data we obtained will provide valuable information for studying evolutionary processes in the Capsicum genus and will assist in the functional analysis of Capsicum mitogenomes.

## 1. Introduction

Chilean peppers (*Capsicum* spp.) have gained widespread popularity as versatile vegetable and spice crops. These peppers were initially discovered in the Americas and were later introduced as substitutes for the highly valued Asian pepper (*Piper nigrum* L.), which led to their rapid spread, first to Europe, and then to Asia and Africa [1]. Today, peppers have become one of the most extensively used plants in culinary dishes [2]. Peppers offer a wide variety of culinary uses, making them highly appreciated for their versatility. They are commonly used as spices or vegetables, imparting a distinctive color and flavor to various dishes, pastes, and sauces [3]. Furthermore, peppers can be incorporated into recipes in both their ripe and unripe forms, though the preference among European consumers tends to lean towards ripe peppers [4]. The diverse range of uses and flavors that peppers provide contribute to their ubiquity in global cuisine. They have become a staple ingredient in many culinary traditions, showcasing the rich cultural significance associated with these vibrant fruits. With their vibrant colors, unique tastes, and versatility, peppers continue to captivate taste buds and add depth to a multitude of gastronomic creations.

Cultivars belonging to the genus *Capsicum* encompass a lesser-known species, *C. pubescens* Ruiz & Pav. (purple and black wrinkled seeds) [5]. This particular species, referred to as rocotos, has been cultivated by the inhabitants of the Andean region for millennia. Rocotos are widely utilized in Andean cuisine and play a fundamental role in the region’s ethnobotanical knowledge. Additionally, despite their limited extent, rocotos are primarily found in the highlands of Central America. Interest and demand for them have recently surged in Europe, notably in Spain, due to the increasing number of South American immigrants. Furthermore, the growing popularity of ‘ethnic’ foods and the proliferation of Andean-style restaurants have created an expanding market for these species in other developed countries [6,7,8].

Mitochondria are complex intracellular organelles, long known as the cellular powerhouse due to their critical role in energy production through oxidative metabolism. Serving as the central metabolic hub, mitochondria quickly respond to various environmental cues and metabolic changes to meet the bioenergetic demands of cells. This phenomenon has recently been defined as mitochondrial plasticity [9]. With their high plasticity, mitochondria constitute a dynamic network of signaling organelles that fulfill important multifunctional roles in cell metabolism, proliferation, and survival [10]. Notably, several preclinical and clinical studies have demonstrated that metabolic and/or genetic mitochondrial alterations contribute to the development of various diseases, including cancer [11,12].

Plant mitochondrial (mt) genomes are typically over 1000 times larger than animal mt genomes and exhibit more complex structures due to frequent recombination [13]. The landscape of plant mt genomes is shaped by various factors, including a gene-rich composition with abundant introns, the capacity to include and integrate sequences transferred within the cell from the chloroplast and nucleus, and frequent horizontal gene transfer events from foreign donors [14,15,16,17]. While research on *C. pubescens* is extensive, studies on its mt genome are relatively scarce compared to other plants in the *Capsicum* genus. The few data that we could find in NCBI were those of the complete mitochondrial genome of the Solanaceae family, especially the genus *Capsicum*. Thus, the studies in this area are inadequate.

In this study, we assembled and annotated the complete mt genome of *C. pubescens*. We investigated several aspects of its genome, including characteristics, codon usage, RNA editing sites, repeat sequences, selective pressure, gene clusters, and phylogenetic relationships. Furthermore, we compared it with other plant mt genomes. The data we obtained will provide valuable information for studying evolutionary processes in the *Capsicum* genus and will assist in the functional analysis of Capsicum mitogenomes. This article is organized as follows: Section 2 introduces the materials used in the study and the methods and parameters for DNA extraction, sequencing, assembly, and annotation, as well as specific methods for codon usage, RNA editing sites, repeat sequences, selective pressure, gene clusters, and phylogenetic analysis. Section 3.1 discusses the mitochondrial genome characteristics of *C. pubescens*. Section 3.2 presents the comparative analysis of *C. pubescens* with seven other plants, including *C. annuum* and *C. chinense* from the Capsicum genus.

## 2. Materials and Methods

### 2.1. Plant Material and DNA Extraction

Germplasm resources were sourced from the Jiangsu Academy of Agricultural Sciences. The sample was collected from Guiyang City, Guizhou Province, China (26.5° N, 106.66° E), and the collected sample was stored in the laboratory of the Ministry of Education at Nanning Normal University (sample code NN202010). Total genomic DNA was extracted from the leaves using a modified CTAB method [18]. Agarose gel electrophoresis and Qubit (Thermo Fisher Scientific, Waltham, MA, USA) were utilized to assess the quality of the DNA fragments.

### 2.2. DNA Sequencing, Genome Assembly, and Annotation

High-quality genomic DNA was extracted from leaves using a modified CTAB method [18]. Library construction was performed using an SQK-LSK109 genomic sequencing kit (ONT, Oxford, UK), and DNA sequencing was conducted on the Nanopore PromethION sequencing platform (Oxford Nanodrop Technologies, Wilmington, DE, USA), resulting in 14.28 Gb of raw data with an average read size of 7707 bp. The obtained data underwent filtering and were postprocessed using NanoFilt and NanoPlot in Nanopack [19]. A total of 13 GB of data was derived from 1.66 million reads, with an average read size of 7.82 KB. Libraries with a mean fragment length of 350 bp were constructed using the Nextera XT DNA Library Preparation Kit (Illumina, San Diego, CA, USA). Subsequently, sequencing was conducted on the Illumina Novaseq 6000 platform, generating 9.93 Gb of raw sequence data. After processing using the NGS QC Tool Kit (v. 2.3.3) [20], 9.88 Gb of data was generated from 33.09 million reads. Initially, we obtained a computationally efficient rough assembly using Miniasm [21]. Adapter sequences were trimmed using Porechop (v. 0.2.4) [22], and the resulting assembly was polished using Racon [23]. We identified contigs with similarity to the *Capsicum annuum* (NCBI reference sequence: KJ865410) mitochondrial genome using Bandage (v. 0.9.0) [24]. We retained contigs with at least one alignment of ≥7 kb to the *Gossypium arboretum* mitochondrion using BlastN [25]. Subsequently, we aligned the Nanopore reads to our draft *C. pubescens* assembly using Minimap2 (v. 2.24) [26]. Aligned reads were isolated and assembled de novo, first using Unicycler and later using Flye (v. 2.9.3) [27]. The final genome sequence was obtained via polishing with Pilon using Illumina Novaseq 6000 sequencing reads. The mitochondrial genomes were annotated using BlastN and Mitofy [14], and angiosperm mitochondrial genes were used to query sequences in the NCBI database (https://www.blast.ncbi.nlm.nih.gov (accessed on 10 October 2023)). The tRNA genes were identified using the tRNA scan-SE software (v. 2.0.12) (http://lowelab.ucsc.edu/tRNAscan-SE/ (accessed on 10 October 2023)). The circular map and syntenic gene cluster maps of the plant mitochondrial genomes were generated using OGDRAW (v. 1.3.1) (http://chlorobox.mpimp-golm.mpg.de/ (accessed on 15 October 2023)) [28]. To identify mitochondrial synteny blocks of the *C. pubesens* Ruiz & Pav compared with representative species, 7 pairs of mitogenomes were aligned using Mauve v. 2.3.1. The amount of shared mtDNA was determined using BLAST with an e-value of 10 × 10^−5^.

### 2.3. Codon Usage, Selective Pressure, RNA Editing Prediction, and Repeat Sequence Identification

The frequency of codon usage and relative synonymous codon usage (RSCU) were analyzed for all protein-coding genes (PCGs) using codonW (v. 1.4.4) [29]. To assess the selective pressure on PCGs, we calculated the nonsynonymous (Ka) and synonymous (Ks) substitution rates between *C. pubescens* and seven other higher plants for PCG. Orthologous gene pairs were aligned and formatted using ParaAT (v. 2.0) with default parameters [30]. Ka, Ks, and Ka/Ks values were calculated using KaKs_Calculator (v. 2.0) based on the YN method [31]. To validate the Ka and Ks values, Fisher’s exact test was performed.

The RNA editing sites in the PCGs of *C. pubescens* and the other seven mitochondrial genomes were predicted using the online PREP-Mt (predictive RNA editors for plants) suite of servers (http://prep.unl.edu/ (accessed on 20 October 2023)) with a cut-off score of C = 0.2 [32]. 

Three types of repeats, SSRs, tandem repeats, and dispersed repeats, were detected in *C. pubescens*. SSRs were identified using the web-based microsatellite identification tool MISA-web (https://webblast.ipk-gatersleben.de/misa/ (accessed on 20 October 2023)), with a motif size of 1–6 nucleotides and thresholds of 8, 4, 3, 3, 3, and 3, respectively. The minimum distance between two SSRs was set to 100 bp. Tandem repeats were identified by running the web-based Tandem Repeats Finder (https://tandem.bu.edu/trf/trf.html (accessed on 20 October 2023)), with alignment parameters set to 2, 7, and 7 for matches, mismatches, and indels, respectively. Dispersed long repeats were determined by running the REPuter program (https://bibiserv.cebitec.uni-bielefeld.de/reputer (accessed on 20 October 2023)) with a minimum repeat size of 30 bp and a 90% sequence identity [33]. The number of dispersed repeats was calculated for six size intervals (30–49, 50–69, 70–99, 100–149, 150–199, and ≥200 bp).

### 2.4. Phylogenetic Analysis

Phylogenetic trees were constructed using the 25 mt genome sequences of the selected species downloaded from the NCBI Organelle Genome Resources database (http://www.ncbi.nlm.nih.gov/genome/organelle/ (accessed on 25 October 2023)). From these 25 species (Appendix A), we selected 12 homologous protein-coding genes (PCGs). To align the sequences, we utilized MAFFT (v. 7.429) with the FFT-NS-2 strategy and a model finder to select GTR + F + R2 [34,35]. The phylogenetic tree was constructed using IQTREE (v. 1.6), employing maximum likelihood methods with 1000 bootstrap replicates. *Oryza sativa indica* (NC_007886) was used as the outgroup [36].

## 3. Results and Discussion

This section is divided into two parts. The first part presents the features of the mitochondrial genome of *C. pubescens*, including its structure, gene content, repeat sequences, and codon usage. Our focus is solely on *C. pubescens* in this analysis. In the second part, we conduct a comparative analysis and discussion of the mitochondrial genome of *C. pubescens,* concerning the mitochondrial genomes of other species.

### 3.1. Genomic Features of the C. pubescens Mitochondrial Genome

#### 3.1.1. Mitochondrial Structure and Gene Content

The mitochondrial genome of *C. pubescens*, revealed through the assembly process, exhibits a more complex structure compared to the typical singular circular genomes found in common plants. Similar to other peppers, such as *C. annuum* var *glabriusculum* [37], it consists of four multiple circular structures. The total size of the mitochondrial genome was 454,275 bp, with each of the four circles measuring 60,995 bp, 87,340 bp, 212,675 bp, and 93,265 bp, respectively (Figure 1A–D). The overall GC content was 44.28%, with the GC contents of the four circles being 44.87%, 44.22%, 44.65%, and 43.11%, respectively (Appendix A). A total of 60 unique genes were detected in the *C. pubescens* mitogenome, including 39 PCGs, 18 tRNA genes, and three rRNA genes. The PCGs comprise only a negligible portion of the genome, totaling 33,807 bp. Among the PCGs, the nucleotide composition was A: 26.4%, C: 21.0%, G: 21.0%, and T: 31.6%, resulting in an overall GC content of 42.0%. All PCGs were found to have a single copy, except for the *cox1* gene, which had two copies. Additionally, 10 genes (*nad7*, *nad1*, *nad4*, *ccmFC*, *rpl2*, *rps10*, *rps3*, *nad2*, *nad5*, and *cox2*) contained one or more introns. In terms of RNA, there were three rRNA genes (*rrn5*, *rns*, and *rnl*) with a single copy, and six tRNA genes (*trnM*-*CAT*, *trnY*-*GTA*, *trnE*-*TTC*, *trnL*-*CAA*, *trnP*-*TGG*, and *trnD*-*GTC*) with two or four copies, with *trnM*-*CAT* having four copies (Table 1 and Appendix A). Compared with *C. annuum* L., there are a total of 35 protein-coding genes in *C. pubescens*, with differences observed in ribosomal proteins (SSUs). *C. pubescens* possesses unique *rps1* and *rps7* genes [38].

To investigate the mitochondrial genomic rearrangement of *C. pubescens*, the shared DNA amount between *C. pubescens* and seven other plant species was analyzed. As shown in Figure 2, compared with other plants, the mitochondrial genome sequence of *C. pubescens* exhibited a partially high level of homology and shared DNA. For instance, in the 454 kb mitochondrial genome of *C. pubescens*, 85% (386 kb) was homologous to the mitochondrial genome of *C. annuum* cultivar *Jeju*, and 91% (412 kb) was homologous to the mitochondrial genome of *Capsicum chinense*, indicating a high degree of homology among these three pepper varieties. In contrast, compared with other species, 53% (243 kb) was shared between *C. pubescens* and *Solanum sisymbriifolium*, 46% (210 kb) was shared between *C. pubescens* and *Nicotiana attenuata*, 31% (139 kb) was shared between *C. pubescens* and *Capsicum rupicola*, 30% (134 kb) was shared between *C. pubescens* and *Osmanthus fragrans*, and only 26% (117 kb) was shared between *C. pubescens* and *Peganum harmala*.

#### 3.1.2. Codon Usage Analysis of PCGs

In the *C. pubescens* mt genome, the majority of the PCGs utilize ATG as the start codon. However, *mttB* starts with TTG, and the start codon for rps10 is ACG (presumably due to C-to-U RNA editing on the second site) (Appendix A). The start codon for rps7 was undetermined. Three types of stop codons were observed in the PCGs: (1) TAA (20 genes: *atp1*, *atp8*, *cox1* (two copies), *cox2*, *nad1*, *nad2, nad3*, *nad4L*, *nad5*, *nad6*, *nad9*, *rpl2*, *rpl5*, *rpl10*, *rpl16*, *rps4*, *rps7*, *rps19*, and *sdh4*), (2) TAG (six genes: *atp4*, *atp9*, *ccmFn*, *mttB*, *nad7*, and *rps3*), and (3) TGA (13 genes: *atp6*, *ccmB*, *ccmC*, *ccmFc*, *cob*, *cox3*, *matR*, *nad4*, *rps1*, *rps10*, *rps12, rps13*, and *sdh3*). The codon usage analysis (Appendix A) revealed that leucine (Leu) is the most frequently used amino acid residue, while cysteine (Cys) is the least used in *C. pubescens* mitochondrial proteins. Moreover, the RSCU analysis of the *C. pubescens* mt genome is shown in Figure 3. It is observed that for the first codon of all 23 amino acids, six codons have RSCU values lower than one, two codons have RSCU values of one, and the remaining codons have RSCU values higher than one. However, for the second codon of 21 amino acids, six codons have RSCU values higher than one, while the others have RSCU values lower than one, except for one codon with an RSCU value of one. Only nine amino acids had a third codon, and eight had a fourth codon. In the fourth codon, the RSCU values were not higher than one, and for the third codon, they were not lower than one, except for AUC (RSCU = 0.82). The codon usage pattern of the *C. pubescens* mt genome is similar to that of most plants’ mt genomes, as discussed in the subsequent section.

#### 3.1.3. Repeat Sequences Analysis

In this study, we detected SSRs, tandem repeats, and dispersed repeats in *C. pubescens*. In the whole genome, 359 intact SSRs were identified, including mono-, di-, tri-, tetra-, penta-, and hexanucleotide repeats (Appendix A). Of these, Circle 1 contained 43 SSRs, Circle 2 had 81 SSRs, Circle 3 had 160 SSRs, and Circle 4 had 75 SSRs. The abundance of SSRs exhibited a negative correlation with complexity, with the following quantitative relationship: mono- > di- > tri- > tetra- > penta- > hexanucleotide repeats. While mononucleotide repeats are generally the most common type, in the *C. pubescens* mt genome, dinucleotide repeats were found to be more prevalent than other repeat types (154 SSRs), with mononucleotide repeats being the second most numerous (143 SSRs). These two types together accounted for 82.79% of the total SSRs, while the remaining four types accounted for only 17.21%. Hexanucleotide repeats were the least numerous, with only two repeats found in the entire mt genome. Furthermore, tetranucleotide repeats were more prevalent than trinucleotide repeats. Sidewise, among mononucleotide repeats, those with eight repeats were the most common (76 SSRs), with no repeats longer than 15 or shorter than eight. The longest mononucleotide repeat, spanning 15 repeats, was localized in Circle 2. In dinucleotide repeats, the most common type was those with four repeats (134 SSRs), with repeat numbers greater than seven or less than four. Trinucleotide repeats were represented by only two types: those with four repeats (eight SSRs) and those with five repeats (three SSRs). Tetra-, penta-, and hexanucleotides consisted of only one type, that is, specifically three repeats. 

Tandem repeats were also detected: 34 tandem repeats with lengths ranging from 7 to 39 bp were detected in the *C. pubescens* mt genome, with a collective copy number of 91.1 (Appendix A). Most of the tandem repeats were localized in Circle 3, whereas Circle 1 exhibited the fewest tandem repeats. This observation could potentially be attributed to the smaller size of Circle 1 in the *C. pubescens* mt genome, with Circle 3 being the largest. Furthermore, among these 34 tandem repeats, only one was positioned in a coding region (*rpl16* in Circle 4), two repeats were localized in the intron region (*rpl2* in Circle 3), and the remaining repeats were found within intergenic spacers.

In addition to SSRs and tandem repeats, 58 dispersed repeats with lengths ≥ 30 bp were identified in the *C. pubescens* mt genome. Two types of long repeats were identified, comprising 38 forward repeats and 20 palindromic repeats. However, no inverse repeats or complementary repeats were detected. The majority of the repeats (67.2%) ranged from 30 to 49 bp in length, while four repeats exceeded 100 bp, with a single repeat surpassing 1 kb in length (Cir3: 3833 bp).

### 3.2. Competitive Analysis of C. pubescens Mt Genomes with Other Angiosperms

In this section, we conduct a comparative analysis of the mt genome of *C. pubescens* with the mt genomes of seven species from three different families. Our selection includes two species from the Capsicum genus (*C. annuum* and *C. chinense*), two species from Solanaceae (*N. attenuate* and *S. sisymbriifolium*), two species from Oleaceae (*C. rupicola* and *O. fragrans*), and one species from Nitrariaceae (*P. harmala*). In the subsequent sections, we compare these genomes based on codon usage, repeat sequences, gene clusters, selective pressure, and phylogenetic analysis.

#### 3.2.1. Codon Usage Analysis and RNA Editing Prediction in PCGs

In Section 3.1.2, we presented the codons found in the mt genome of *C. pubescens*. However, our analysis includes not only *C. pubescens* but also all the other species examined in this study. Referring to the previous studies, we selected representative species of *C. annuum*, representative species belonging to the Solanaceae family, and representative monocotyledonous and dicotyledonous species with a complete mitochondrial genome as the available reports [38,39]. The range of codons observed in these seven species is from 26,769 to 37,770. Among these species, *O. fragrans* has the highest number of codons (37,770), while *P. harmala* has the lowest number (26,769). The genus *Capsicum* exhibits significant variation in the number of codons, ranging from 31,362 to 33,690, with a difference of 2328 codons. Detailed information is presented in Figure 3 and Appendix A. Our codon usage analysis revealed that Leu is the most frequently used amino acid residue in all eight species. In addition, we found that the relative synonymous codon usage (RSCU) proportions of the various amino acids in *C. pubescens* and *Phaseolus vulgaris* were very similar, but there were differences in the codon preferences for encoding amino acids [40]. However, there are variations in the least-used amino acid residues among the species. Cys is the least-used amino acid residue in *C. pubescens*, *C. annuum*, *C. rupicola*, *N. attenuata*, *O. fragrans*, and *P. harmala* mitochondrial proteins. On the other hand, tryptophan (Trp) is the least-used amino acid residue in the *C. chinense* and *S. sisymbriifolium* mitochondrial proteins. The codon preferences among the three Capsicum species were generally similar, with little variation in the frequency of codon usage.

We proceeded to predict the RNA editing sites in PCGs and identified 501 sites of RNA editing in 39 PCGs of *C. pubescens*. However, in Circle 1 and Circle 3, we observed eight repeated sites in the *cox1* gene, resulting in a total of 493 RNA editing sites (Figure 4 and Appendix A). The range of RNA editing sites across the eight mt genomes of species was between 430 and 493, with *C. pubescens* still exhibiting the highest number of RNA editing sites among all the species, while *P. harmala* had the fewest, with 430 sites. In particular, the *ccmB* gene harbored the highest number of RNA editing sites in all species, except *C. rupicola* and *O. fragrans*, which had the highest number of RNA editing sites in the *nad4* gene. Moreover, *cox1* displayed a significantly greater number of RNA editing sites in *C. pubescens* compared to other species, while *C. annuum* and *C. chinense* exhibited much fewer sites than *C. pubescens*. Based on Appendix A, we identified all C-to-U RNA editing sites in the PCGs. Out of these, 155 sites were predicted at the first base position of the codon, 313 sites were found at the second position, and the remaining 22 sites were observed at both positions. Additionally, *C. pubescens* had two RNA editing sites in the rps7 gene, which were absent in the other two peppers. *C. chinense* had a slightly lower number of RNA editing sites in the atp1 gene compared to other Capsicum species. *C. annuum* lacked RNA editing sites in the atp4 and rps1 genes, whereas other Capsicum species possessed them.

#### 3.2.2. Dispersed Repeats Analysis

Dispersed repeat sequences are thought to have significant implications in the mitochondrial genome. These long repetitive sequences have also proven valuable as markers for exploring plant evolution, comparative genomics, and phylogenetics. The total number of repeats varied between 58 bp in *C. pubescens* and 218 bp in *C. annuum* (Figure 5 and Table 2). In most species, the majority of repeats ranged from 30 to 49 bp in length. However, in *O. fragrans*, they predominantly fell within the range of 100 to 149 bp. Except for *O. fragrans*, the smallest number of repeats ranged from 150 to 199 bp in length. Among the species, there were 22 large repeats (>1 kb), ranging from 1074 bp in *S. sisymbriifolium* to 17,243 bp in *C. annuum*, including one for *C. pubescens*, five for *C. annuum*, two for *C. chinense*, three for *N. attenuata*, seven for *P. harmala,* and four for *S. sisymbriifolium*. Our investigation revealed the presence of dispersed repeats across eight angiosperm mt genomes. However, these repeats exhibited poor conservation across species, even within the same genus. It is worth noting that the Capsicum genus exhibited significant variation in this aspect. In particular, *C. pubescens* showed notable differences, such as a lower number of repeat sequences, with lengths ranging from 30 to 49, and 70 to 99 base pairs compared to other Capsicum species. Moreover, the overall quantity of dispersed repeat sequences in *C. pubescens* followed this trend. This may be attributed to differences in their mitochondrial genome structure. While both *C. chinense* and *C. pubescens* have multiple circular structures, other species featured single-circle structures. This also indicated that *C. pubescens* exhibited greater variation, similar to the results of the phylogenetic analysis below.

#### 3.2.3. Selective Pressure Analysis

One approach to evaluating the selective pressure driving protein evolution is to compare the rates of synonymous and non-synonymous nucleotide substitutions. Ks represents the estimated number of synonymous changes per synonymous site, reflecting the rate of neutral amino acid evolution. Conversely, Ka signifies the number of non-synonymous substitutions per non-synonymous site. Under neutral protein-level evolution, Ka should be equal to Ks, resulting in a Ka/Ks ratio of 1. A Ka/Ks ratio < 1 suggests purifying or stabilizing selection, indicating resistance to change. Conversely, a ratio of >1 implies positive or Darwinian selection, promoting change. In this study, we analyzed 39 PCGs from *C. pubescens* and compared them with those of seven other species. Notably, by considering the number of PCGs with nonzero Ka/Ks values, we could preliminarily infer the phylogenetic relationships between the species. Although this method may not provide precise information on the level of relatedness, it offers evidence of their degree of divergence. The Ka/Ks ratio serves as an indicator of protein evolution or divergence among species. As shown in Figure 6, four PCGs exhibited nonzero values of Ka/Ks, with *atp6*, *nad5*, and *sdh3* displaying ratios > 1 between *C. pubescens* and *C. annuum*, indicating that these genes may have experienced positive selection since their last common ancestor. Nine PCGs had nonzero Ka/Ks values, but only *rps10* had a Ka/Ks ratio > 1 between *C. pubescens* and *C. chinense*. There were differences in the comparison between these two pairs, indicating that *C. pubescens* had different evolutionary selection directions from these two Capsicum species. Moreover, when comparing *C. pubescens* with *S. sisymbriifolium*, 17 PCGs showed nonzero values of Ka/Ks, indicating potential positive selection in *ccmFc*, *mttB*, and *matR* after divergence. When comparing *C. pubescens* with *N. attenuata*, 29 PCGs displayed evolution, and *ccmFc*, *matR*, *mttB*, *rps10*, and *sdh3* may have undergone positive selection. These species belong to the same family. Additionally, we compared species from different families such as *C. rupicola*, *O. fragrans*, and *P. harmala* to *C. pubescens*. Using *P. harmala*, 29 PCGs exhibited potential positive selection, including *atp4*, *ccmC*, and *rpl16*. Both *O. fragrans* and *C. rupicola* had the same number of PCGs with nonzero Ka/Ks values, with 37 mutated PCGs, all being identical. However, five genes (*atp4*, *nad1*, *ccmC*, *mttB*, and *rpl2*) in *C. rupicola* may have undergone positive selection, while four genes (*atp4*, *ccmC*, *mttB*, and *rpl2*) were observed in *O. fragrans*. Based on the findings from Section 3.2.1, we found that the mt genome of *P. harmala* contained 30 PCGs, suggesting a greater difference between *C. pubescens* and *P. harmala* compared to *C. pubescens* and *N. attenuata*.

#### 3.2.4. Gene Clusters and Phylogenetic Analysis

Based on the information presented in Appendix A and Figure 7, three gene clusters ((*cob*)-(*rps14*)-*rpl5*, *trnC*-*trnN*-*trnY*-*nad2*, and (*rps19*)-*rps3*-*rpl16*-(*cox2*)) were highly conserved across all species. Furthermore, eight gene clusters (*trnS*-*trnF*-*trnP*, *rrn18*-*rrn5*, *rps13*-*nad1*, *nad9*-(*trn P*)-*trnW*, *rps12*-*nad3*, *nad1*-*matR*, *sdh3*-*nad2*, and *ccmC*-*trnL*/*trnM*) were widely distributed in most species, except for *P. harmala*. The presence of the gene cluster (*atp8*)-*cox3*-*sdh4* was observed in all species except *S. sisymbriifoliume*. The distribution of the gene cluster *rrn26*-*trnM* was widespread in most species, except for *O. fragrans* and *C. rupicola*. The presence of the gene cluster *rps10*-*cox1* was found in *C. pubescens*, *S. sisymbriifolium*, and *N. attenuata*, but it was scattered in other mt genomes. Conversely, the gene cluster *atp4*-*nad4L* was scattered in *C. annuum*, *S. sisymbriifolium*, and *C. rupicola*, but present in other mt genomes. The gene cluster *rpl2*-*rpl10* was widely distributed in most species, except for *O. fragrans*, *C. rupicola*, and *P. harmala*. Lastly, the gene cluster *nad6*-*rps4* was present in *S. sisymbriifolium*, *N. attenuata*, *O. fragrans*, and *C. rupicola*, but scattered in other mt genomes.

Comparing the gene clusters by family reveals intriguing observations. Among the five Solanaceae species, differences only occurred in the gene clusters *rps10*-*cox1*, (*atp8*)-*cox3*-*sdh4*, *atp4*-*nad4L*, and *nad6*-*rps4*, with no other variations detected. Similarly, among the three Capsicum species, variations were found solely in the *rps10*-*cox1* and *atp4*-*nad4L* gene clusters. Furthermore, only one difference was observed in the gene cluster *atp4*-*nad4L* between the two Oleaceae species. This suggests that a closer relationship exists between species exhibiting fewer differences in gene clusters; however, further evidence is necessary to support this conclusion.

With rapid advancements in genomic sequencing and molecular biology, similar clustered gene organization has been observed in the mitochondria of various green plants, ranging from charophyte algae to angiosperms. Due to factors such as repeat-mediated homologous recombination, sequence duplications, genome expansion and shrinkage, and the incorporation of foreign DNA material, plant mitogenomes display high variability in their genome structure. Although ancestral ribosomal gene clusters often disintegrate into smaller clusters of no more than four genes, some gene clusters remain conserved in large-scale phylogenetic analyses. However, gene organization typically differs significantly between plant mt genomes. Generally, gene orders and clusters tend to be more similar among species with closer relationships. In contrast, significant variations in gene orders and clusters can be observed between different families, as the clusters are prone to disruption through genome recombination. This may explain why multiple recombination events can result in the generation of similar syntenic gene clusters, leading to substantial differences in gene order across plant mt genomes.

Simultaneously, we constructed a phylogenetic tree to further analyze the relationships between the species. As shown in Figure 8, the bootstrap values of each node were all supported by over 60%, with 13 nodes receiving 100% support. The ML phylogenetic tree strongly supports the notion that *C. pubescens* is evolutionarily close to *C. annuum* and *C. chinense*. *C. annuum* and *C. chinense* have a closer relationship to each other than to *C. pubescens*. At the genus level, Solanum was found to have a closer relationship with Capsicum than with Nicotiana. In addition, this study also selected other models to construct the phylogenetic tree, and the results were similar (Appendix A). Furthermore, Peganum exhibited the most distant relationship with any other species included in this study. It can also be observed in Appendix A that significant differences occurred in the distribution of gene clusters in the mt genome of *P. harmala*.

## 4. Conclusions

We discovered that the mt genome of the *C. pubescens* consists of four circular structures and has a length of 454,275 bp. It encompasses 39 PCGs, three rRNAs, and 18 tRNAs. Among the eight species analyzed, *C. pubescens* had the highest number of RNA editing sites in PCGs, but not the highest number of codons. *C. pubescens* exhibited the fewest repeat sequences, likely due to the presence of multiple circular structures. Furthermore, our findings from selective pressure analysis, gene cluster comparisons, and phylogenetic analysis indicated a close relationship between *C. pubescens*, *C. annuum*, and *C. chinense*. Notably, *C. annuum* and *C. chinense* were found to be more closely related to each other than to *C. pubescens*. At the genus level, Solanum displayed a closer relationship with Capsicum than with Nicotiana. Moreover, the results from the selection pressure analysis and gene cluster trends supported and provided evidence for the phylogenetic analysis results. The sequencing of the *C. pubescens* mt genome enhances our understanding of mt genome characteristics and their evolution in angiosperms.

## Figures and Tables

**Figure 1 genes-15-00152-f001:**
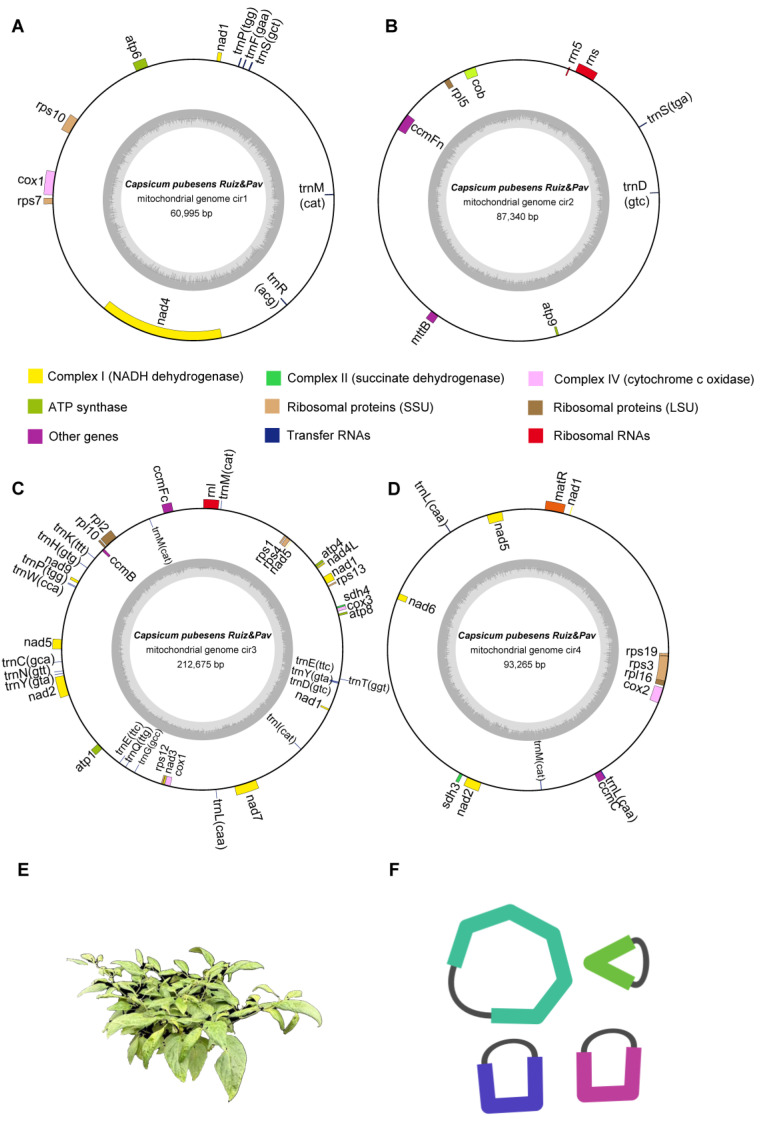
Assembly results of the *C. pubescens* Ruiz & Pav mitogenome. (**A**–**D**) Circular map of the *C. pubescens* Ruiz & Pav mitogenome. (**E**) The morphological diagram of *C. pubescens* Ruiz & Pav. (**F**) The assembly graphs reveal a multipartite genome structure consisting of four circular molecules in the mitochondrial genome of *C. pubescens* Ruiz & Pav.

**Figure 2 genes-15-00152-f002:**
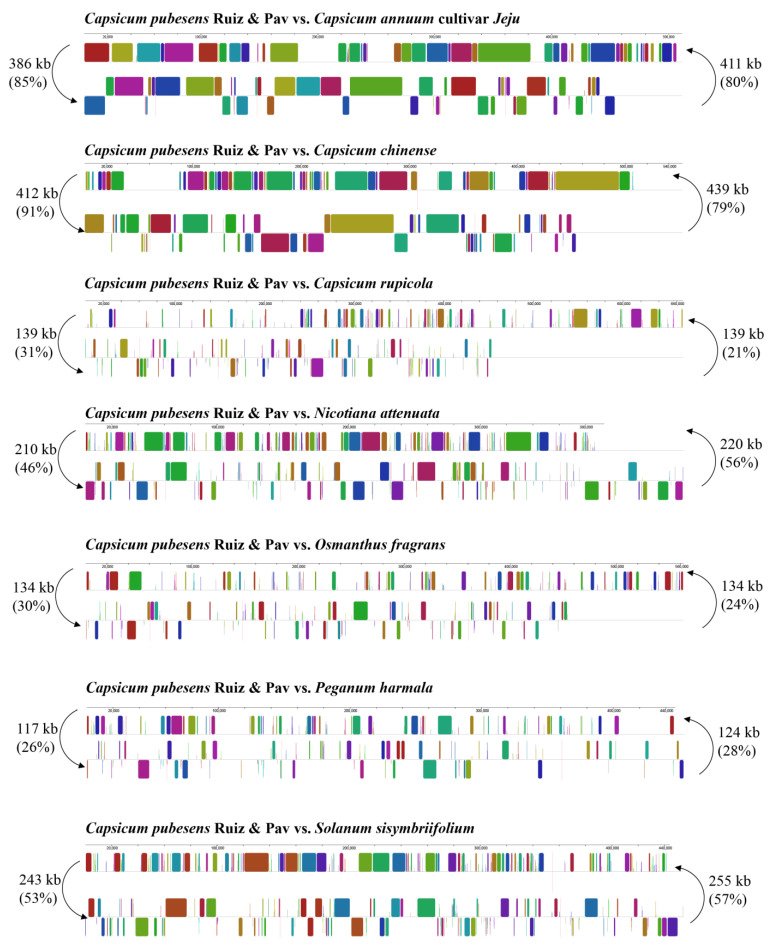
Synteny blocks and shared mtDNA of the *C. pubescens* and seven other plants based on Mauve alignments. Left arrows indicate the amount of shared mtDNA, in kb and percentage, of the *C. pubescens* and one other species, and right arrows present the reciprocal values.

**Figure 3 genes-15-00152-f003:**
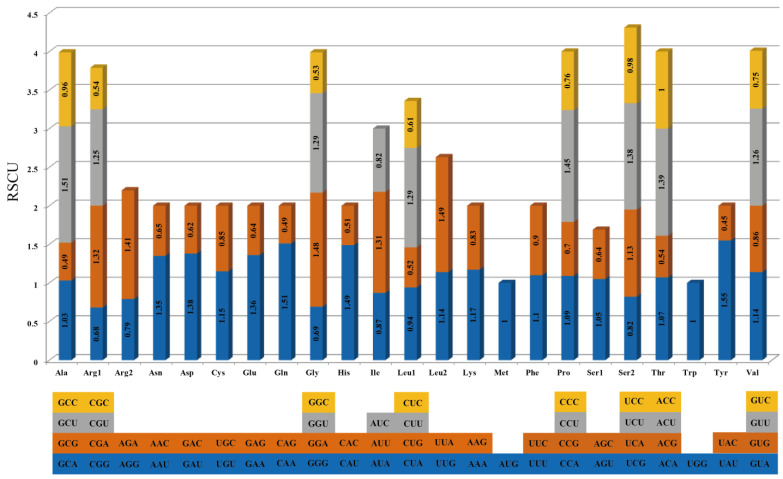
Relative synonymous codon usage (RSCU) of *C. pubescens* mt genome. Codon families are on the X-axis. RSCU values are the number of usage times of a particular codon, relative to the number of times that the codon would be observed for a uniform synonymous codon usage, and different colors in the same column represent different codons coding for the same amino acid.

**Figure 4 genes-15-00152-f004:**
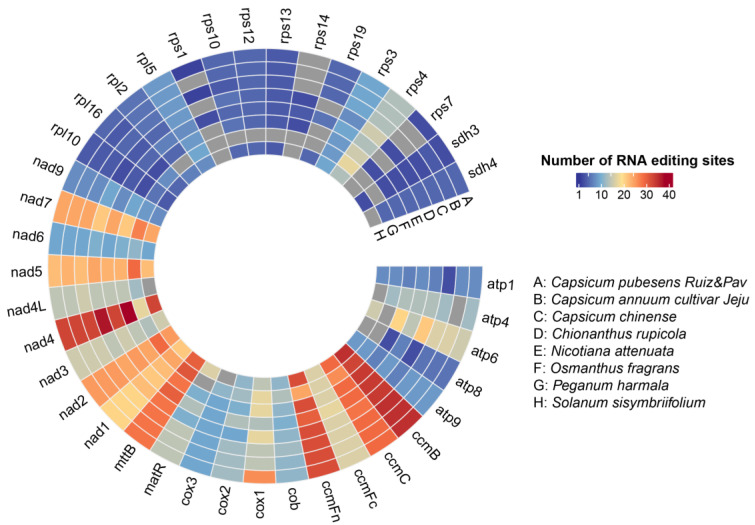
Predicted RNA editing sites in PCGs of *C. pubescens* mt genome with seven other angiosperms.

**Figure 5 genes-15-00152-f005:**
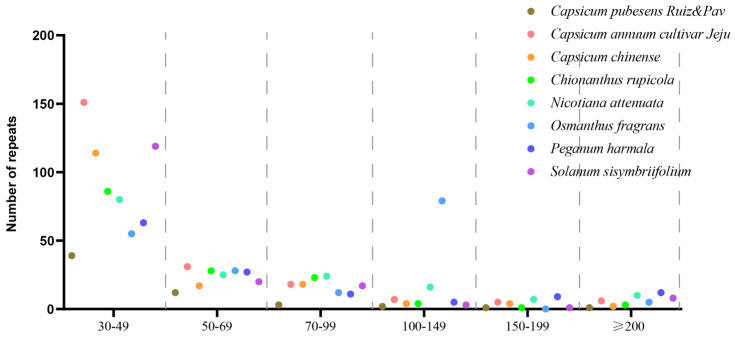
Frequency distribution of dispersed repeats in the *C. pubescens* mt genome compared with seven other angiosperms.

**Figure 6 genes-15-00152-f006:**
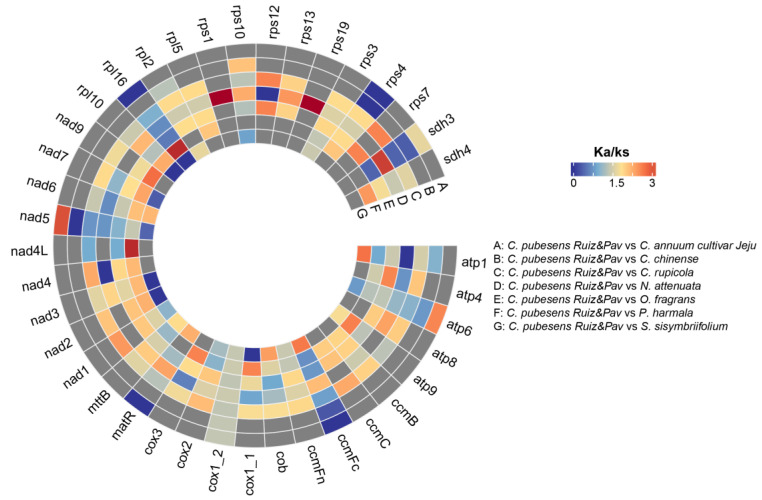
Ka/Ks ratios for PCGs of eight species.

**Figure 7 genes-15-00152-f007:**
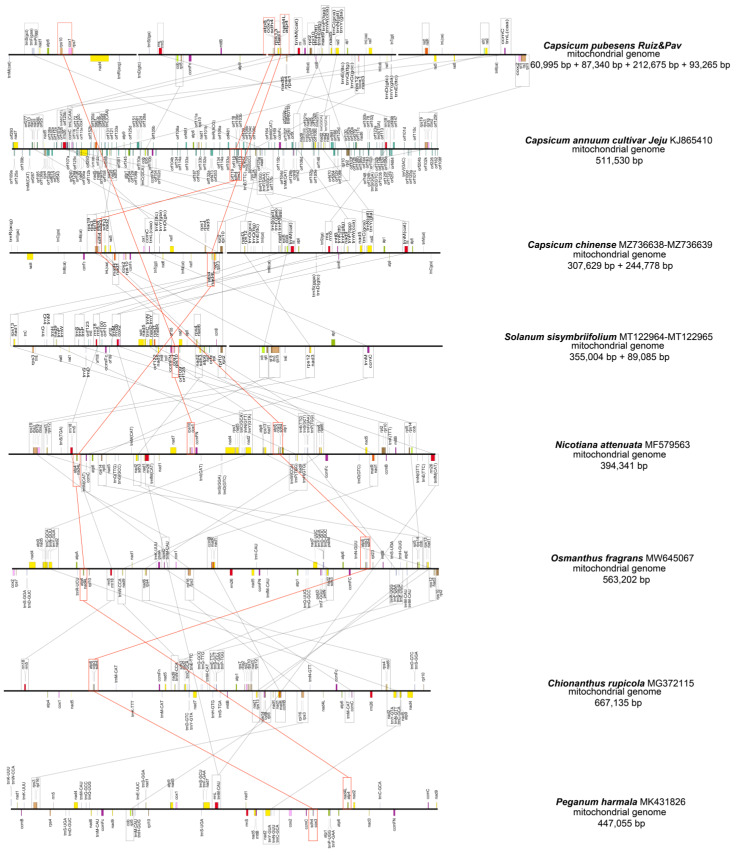
Analysis of conserved gene clusters between the *C. pubescens* mt genome and seven other angiosperms.

**Figure 8 genes-15-00152-f008:**
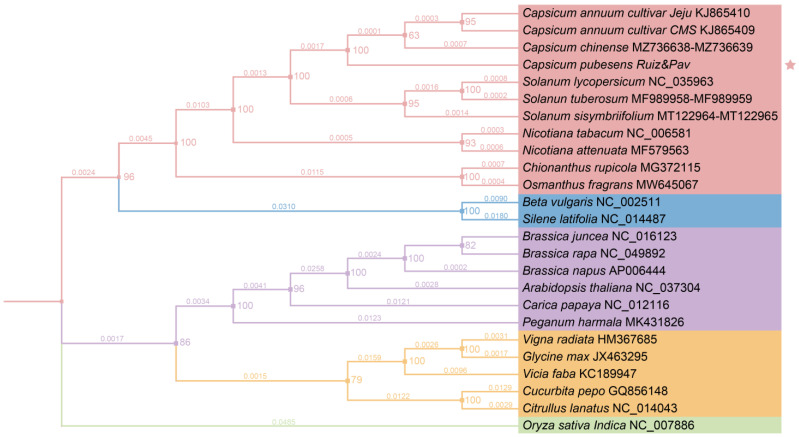
The phylogenetic tree is based on 12 homologous PCGs (atp6, atp9, cob, cox1, mttB, nad3, nad4, nad6, nad7, nad9, rps3, and rps4) in each of the 25 mt genomes. The phylogenetic tree was constructed using IQTREE, employing maximum likelihood methods with 1000 bootstrap replicates. Genes that were present in each sample (respectively, atp6, atp9, cob, cox1, mttB, nad3, nad4, nad6, nad7, nad9, rps3, and rps4) were selected and used in tandem for constructing the phylogenetic tree. Mafft was compared, Gblocks was used for cleaning, and IQTREE was used to construct the ML tree. Base substitution models were detected using IQTREE’s model finder for best model selection. The best nucleic acid substitution model according to the BIC (Bayesian Information Criterion) was GTR + F + R2. Outgroup: *Oryza sativa Indica*, NC_007886. The study categorized species into five groups based on their degree of kinship and distinguished them with different colors. The asterisk indicates the species studied.

**Table 1 genes-15-00152-t001:** Gene content of *C. pubescens* mt genome.

Group of Genes	Gene Name				
	cir 1	cir 2	cir 3	cir 4	Numbers
Complex I (NADH dehydrogenase)	*nad1* ^#^, *nad4*	*-*	*nad1* ^#^, *nad2* ^#^, *nad3*, *nad4L*, *nad5* ^#^, *nad7*, *nad9*	*nad1* ^#^, *nad2* ^#^, *nad5* ^#^, *nad6*	9
Complex II (succinate dehydrogenase)	*-*	*-*	*sdh4*	*sdh3*	2
Complex III (ubiquinol cytochrome c reductase)	*-*	*cob*	*-*	*-*	1
Complex IV (cytochrome c oxidase)	*cox1*	*-*	*cox1*, *cox3*	*cox2*	4
Complex V (ATP synthase)	*atp6*	*atp9*	*atp1*, *atp4*, *atp8*	*-*	5
Cytochrome c biogenesis	*-*	*ccmFn*	*ccmB*, *ccmFc*	*ccmC*	4
Ribosomal protein	*rps7*, *rps10*	*rpl5*	*rps1*, *rps4*, *rps12*, *rps13, rpl2*, *rpl10*	*rps3*, *rps19*, *rpl16*	12
Ribosomal RNA	*-*	*rns*, *rrn5*	*rnl*	*-*	3
Transfer RNA	*trnF*, *trnM*, *trnP*, *trnR*, *trnS*	*trnD*, *trnS*	*trnC*, *trnD*, *trnE*(*2*) *, *trnG*, *trnH*, *trnI*, *trnK*, *trnL*, *trnM*(*2*) *, *trnN*, *trnP*, *trnQ*, *trnT*, *trnW*, *trnY*(*2*) *	*trnL*(*2*) *, *trnM*	28
Others	-	*mttB*	-	*matR*	2
Total	70				70

Multicopy genes are presented with *, and bracketed numbers represent the copy number of each gene. Genes with introns are denoted with #.

**Table 2 genes-15-00152-t002:** Frequency distribution of dispersed repeats in the *C. pubescens* mt genome compared with seven other angiosperms.

Length of Repeats	*C. pubescens* Ruiz & Pav	*C. annuum* cultivar Jeju	*C. chinense*	*C. rupicola*	*N. attenuata*	*O. fragrans*	*P. harmala*	*S.* *sisymbriifolium*
30–49	39	151	114	86	80	55	63	119
50–69	12	31	17	28	25	28	27	20
70–99	3	18	18	23	24	12	11	17
100–149	2	7	4	4	16	79	5	3
150–199	1	5	4	1	7	0	9	1
≥200	1	6	2	3	10	5	12	8
Total	58	218	159	145	162	179	127	168

## Data Availability

The genome sequence data that support the findings in this study are openly available in GenBank of the NCBI at https://www.ncbi.nlm.nih.gov/ (accessed on 30 October 2023) under accession Nos. MZ823411, MZ823412, MZ823413, and MZ823414. The associated BioProject, SRA, and BioSample numbers are PRJNA766458, SRR16091335, SRR16091336, and SAMN21848564, respectively.

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
