# Peer review of "Characteristics and Comparative Analysis of the Special-Structure (Non-Single-Circle) Mitochondrial Genome of Capsicum pubescens Ruiz & Pav"

_genes, 2024, doi:10.3390/genes15020152_

Round 1

Reviewer 1 Report

Comments and Suggestions for Authors

Reviewer’s Comment / Report

The manuscript #genes-2803490 entitled “Characteristics and comparative analysis of special structure (non-single circle) mitochondrial genome of Capsicum pubescens Ruiz & Pav." has been reviewed.

The authors have done significant work on sequencing of the C. pubescens mt genome and comparative analysis of special structure with related species.

There have been several published papers regarding the comparative analysis of Capsicum plastome. This manuscript reveals that the study assembled and annotated the complete mt genome of C. pubescens, which is novelty. The following are some concerns that need to be resolved.

The manuscript concluded that the selective pressure analysis, gene cluster comparisons, and phylogenetic analysis indicated a close relationship between C. pubescens, C. annuum, and C. chinense. Notably, C. annuum and C. chinense were found to be more closely related to each other than to C. pubescens. Which is well known fact that C. pubescens comes under a different clade in many phylogenetic studies in the Capsicum species. However, throughout the manuscript, only the phylogenetic analysis (Fig 7) is somewhat related to this conclusion, the other related species within the study does not provide any valid information and the figures are very poor to understand which needs to be revised.

Author Response

Responds to the Reviewers’ comments:

Reviewer #1:

  1. The manuscript concluded that the selective pressure analysis, gene cluster comparisons, and phylogenetic analysis indicated a close relationship between C. pubescens, C. annuum, and C. chinense. Notably, C. annuum and C. chinense were found to be more closely related to each other than to C. pubescens. Which is well known fact that C. pubescens comes under a different clade in many phylogenetic studies in the Capsicum species. However, throughout the manuscript, only the phylogenetic analysis (Fig 7) is somewhat related to this conclusion, the other related species within the study does not provide any valid information and the figures are very poor to understand which needs to be revised.

Response: Thank you for your valuable comments. We noticed that the content related to these conclusions in the manuscript is insufficient. However, it is important for studying the origin and evolution of Capsicum, as well as related molecular researches. Therefore, we have added a comparative analysis and discussion among Capsicum genera in the manuscript. In addition, we conducted mitochondrial genome collinearity analysis for C. pubescens with seven other species, including C. annuum and C. chinense, to enhance the comparative results among these species. Relevant result descriptions have been added to the manuscript, along with the addition of Figure 2.

Thank you very much for your comments, as it will greatly help improve the quality of the manuscript. The revised parts were marked in red in the revised blinded manuscript. Please see Lines 116-119, 176-178,184-195, 196-199, 281-282, 296-297, 300-303, 319-327, 351-353, 414-415.

We sincerely apologize for the low quality of the figures and tables, which made them difficult to understand. We have made revisions to the figures and tables to better present the relevant research results. Additionally, we have added relevant data information in the supplementary files to facilitate a better understanding. Thank you for bringing this to our attention. Please see Lines 179, 283, 304, 328, 366, 405, 419.

Reviewer 2 Report

Comments and Suggestions for Authors

Manuscript "Characteristics and comparative analysis of special structure (non-single circle) mitochondrial genome of Capsicum pubescens Ruiz & Pav." is very interesting.

General comments:
Authors assembled and annotated the complete mt genome of C. pubescens. Authors investigated several aspects of its genome, including characteristics, codon usage, RNA editing sites, repeat sequences, selective pressure, gene clusters, and phylogenetic relationships. Authors compared it with other plant mitochondrial genomes.

Detailed comments:
The introduction is written correctly.
The description of the material is correct and comprehensive. Unfortunately, the description of the methods used is not complete and does not allow the experiment to be repeated. The Authors did not provide a method for calculating genetic similarity. Measures of similarity are plentiful, and it is necessary to state which one was used in achieving the stated goals of the study.
The quality of the Figure 1 is very poor. Especially A, B, C and D. It is imperative to improve it.
In Table 1, the authors presented groups of genes. The manuscript should be supplemented with a statistical comparative analysis between these groups. Such a comparison seems obvious in this type of study.
Line 200: "In this study, we detected SSRs" - What selection criteria were used? The authors do not mention any statistical methods! Selection of SSRs follows specific restrictive methods, after several assumptions are met. Please complete the manuscript with a description of these criteria and methods as well as relevant results.
The quality of the Figure 3 is very poor.
The quality of the Figure 6 is very poor.
The quality of the Figure 7 is very poor.

Line 11: "L." - not italic

Paper needs major revision.

Author Response

Responds to the Reviewers’ comments:

Reviewer #2:

  1. The description of the material is correct and comprehensive. Unfortunately, the description of the methods used is not complete and does not allow the experiment to be repeated. The Authors did not provide a method for calculating genetic similarity. Measures of similarity are plentiful, and it is necessary to state which one was used in achieving the stated goals of the study.

Response: Thank you for your valuable comments. For the analysis of genetic similarity, we chose to use the method of selective pressure analysis (Ka/Ks) to analyze the relevant content. The specific method details can be found in Section 2.3, Lines 122-128 of the Materials and Methods section. If you have any questions, please feel free to communicate with me at any time. Thank you for your guidance.

  1. The quality of the Figure 1 is very poor. Especially A, B, C and D. It is imperative to improve it.

Response: Thank you for your valuable comments. We sincerely apologize for the poor quality of Figure 1. We have made revisions to improve Figure 1, making it more expressive and understandable. Please see Lines 179.

  1. In Table 1, the authors presented groups of genes. The manuscript should be supplemented with a statistical comparative analysis between these groups. Such a comparison seems obvious in this type of study.

Response: Thank you for your valuable comments. We have added the relevant data information for these groups to Supplementary Table S9. Please refer to Table S9 in the Supplementary Table. In addition, we have revised Table 1 to make the relevant content more explicit and added statistical data for the corresponding genes. The revised parts were marked in red in the revised blinded manuscript. Please see Lines 201-202.

  1. Line 200: "In this study, we detected SSRs" - What selection criteria were used? The authors do not mention any statistical methods! Selection of SSRs follows specific restrictive methods, after several assumptions are met. Please complete the manuscript with a description of these criteria and methods as well as relevant results.

Response: Thank you for your valuable comments. We have realized that the relevant parameters for SSR analysis, although mentioned in section 2.3 of the Materials and Methods, may not be presented in a well-structured manner, making it difficult to read. Therefore, we have improved the content and formatting of this section to enhance readability. The revised parts were marked in red in the revised blinded manuscript. Please see Lines 122-142.

  1. The quality of the Figure 3 is very poor.

Response: Thank you for your valuable comments. We sincerely apologize for the poor quality of Figure 3. We have made revisions to improve Figure 3, making it more expressive and understandable. Please see Lines 304.

  1. The quality of the Figure 6 is very poor.

Response: Thank you for your valuable comments. We sincerely apologize for the poor quality of Figure 6. We have made revisions to improve Figure 6, making it more expressive and understandable. Please see Lines 405.

  1. The quality of the Figure 7 is very poor.

Response: Thank you for your valuable comments. We sincerely apologize for the poor quality of Figure 7. We have made revisions to improve Figure 7, making it more expressive and understandable. Please see Lines 419.

  1. Line 11: "L." - not italic

Response: Thank you for your valuable comments. We sincerely apologize for this mistake, and we have now made the necessary modifications in the manuscript. The revised parts were marked in red in the revised blinded manuscript. Please see Lines 11.

Reviewer 3 Report

Comments and Suggestions for Authors

Review of the article "Characteristics and comparative analysis of special structure (non-single circle) mitochondrial genome of Capsicum pubescens Ruiz & Pav."

In my opinion, the article is interesting and valuable.

I only have a few points to correct:

1. In the lines 144 - 146 is: "The phylogenetic tree was constructed using IQTREE (v1.6), employing maximum likelihood methods with 1,000 bootstrap replicates"

a) Information about the substitution model used should be added.

b) In the MEGA X program it is possible to set 10,000 bootstrap replications - therefore, it would be worth comparing and checking the obtained results (presented in Figure 7) with the results obtained using the MEGA X program.

2. The quality of some figures needs to be improved, for example the quality of Figure 6 is so bad that practically nothing is visible on it.

3. The information presented in the Abstract and in the Conclusions is partly the same and partly similar, for this reason, the Abstract should emphasize more the novelties presented in the article and present the aim of the work more clearly.

Please also consider moving this part (lines 72-77) from the Introduction to the Abstract accordingly "In this study, we have assembled and annotated the complete mt genome of C. pubescens. We investigated several aspects of its genome, including characteristics, codon usage, RNA editing sites, repeat sequences, selective pressure, gene clusters, and phylogenetic relationships. Furthermore, we compared it with other plant mt genomes. The data we obtained will provide valuable information for studying evolutionary processes in the Capsicum genus and will assist in the functional analysis of Capsicum mitogenomes." and finish the Introduction for example in this way: "This article is organized as follows. In section 2, ... In section 3, ..."

Author Response

Responds to the Reviewers’ comments:

Reviewer #3:

  1. In the lines 144 - 146 is: "The phylogenetic tree was constructed using IQTREE (v1.6), employing maximum likelihood methods with 1,000 bootstrap replicates". a) Information about the substitution model used should be added. b) In the MEGA X program it is possible to set 10,000 bootstrap replications - therefore, it would be worth comparing and checking the obtained results (presented in Figure 7) with the results obtained using the MEGA X program.

Response: Thank you for your valuable comments. To compare and validate the results in Figure 7, we first selected MEGA X for a repeat validation analysis. However, due to unforeseen issues, the software was interrupted multiple times. Nonetheless, we chose alternative model parameters for repeat validation. Specifically, we selected genes present in each sample (namely: atp6, atp9, cob, cox1, mttB, nad3, nad4, nad6, nad7, nad9, rps3, rps4), concatenated them to construct the phylogenetic tree. Alignment was performed using mafft (v7.427), and the ML tree was constructed using RAxML (v8.2.12). The nucleotide substitution model used was GTRGAMMA, and the bootstrap value was set to 10,000. The outgroup used was Oryza sativa Indica NC_007886. We have included the validated results in Supplementary Figure S1. Upon comparison, the results were similar. Relevant result descriptions have been added to the manuscript. The revised parts were marked in red in the revised blinded manuscript. Please see Lines 414-415.

  1. The quality of some figures needs to be improved, for example the quality of Figure 6 is so bad that practically nothing is visible on it.

Response: Thank you for your valuable comments. We sincerely apologize for the poor quality of Figure 6. We have made revisions to improve Figure 6, making it more expressive and understandable. Please see Lines 405. In addition, we have made improvements to Figure 1, Figure 3, Figure 4, Figure 5, and Figure 7. Please see Lines 179, 283, 304, 328, 366, 419.

  1. The information presented in the Abstract and in the Conclusions is partly the same and partly similar, for this reason, the Abstract should emphasize more the novelties presented in the article and present the aim of the work more clearly.

Please also consider moving this part (lines 72-77) from the Introduction to the Abstract accordingly "In this study, we have assembled and annotated the complete mt genome of C. pubescens. We investigated several aspects of its genome, including characteristics, codon usage, RNA editing sites, repeat sequences, selective pressure, gene clusters, and phylogenetic relationships. Furthermore, we compared it with other plant mt genomes. The data we obtained will provide valuable information for studying evolutionary processes in the Capsicum genus and will assist in the functional analysis of Capsicum mitogenomes." and finish the Introduction for example in this way: "This article is organized as follows. In section 2, ... In section 3, ..."

Response: Thank you for your guidance. We have made revisions to the abstract and introduction sections. Once again, we sincerely appreciate your assistance in improving the quality of our manuscript. The revised parts were marked in red in the revised blinded manuscript. Please see Lines 14-19, 73-79.

Round 2

Reviewer 2 Report

Comments and Suggestions for Authors

A very well and solidly revised manuscript. The authors have incorporated all my suggested changes and comments. I recommend publishing the manuscript in the current version.